# Structural Complementarity of Bruton’s Tyrosine Kinase and Its Inhibitors for Implication in B-Cell Malignancies and Autoimmune Diseases

**DOI:** 10.3390/ph16030400

**Published:** 2023-03-07

**Authors:** Asim Najmi, Neelaveni Thangavel, Anugeetha Thacheril Mohanan, Marwa Qadri, Mohammed Albratty, Safeena Eranhiyil Ashraf, Safaa Fathy Saleh, Maryam Nayeem, Syam Mohan

**Affiliations:** 1Department of Pharmaceutical Chemistry and Pharmacognosy, College of Pharmacy, Jazan University, P.O. Box 114, Jazan 45142, Saudi Arabia; 2Department of Pharmacology, College of Pharmacy, Jazan University, P.O. Box 114, Jazan 45142, Saudi Arabia; 3Medical Research Center, Jazan University, P.O. Box 114, Jazan 45142, Saudi Arabia; 4Department of Pharmaceutical Analytical Chemistry, Faculty of Pharmacy, Fayoum University, Fayoum 63514, Egypt; 5Substance Abuse and Research Centre, Jazan University, P.O. Box 114, Jazan 45142, Saudi Arabia; 6School of Health Sciences, University of Petroleum and Energy Studies, Dehradun 248007, India; 7Center for Transdisciplinary Research, Department of Pharmacology, Saveetha Dental College, Saveetha Institute of Medical and Technical Science, Saveetha University, Chennai 602117, India

**Keywords:** autoimmune diseases, Bruton’s tyrosine kinase (BTK), B-cell malignancies, kinase domain conformation, covalent inhibitors, non-covalent inhibitors, protein data bank

## Abstract

Bruton’s tyrosine kinase (BTK) is a critical component in B-cell receptor (BCR) signaling and is also expressed in haematogenic and innate immune cells. Inhibition of BTK hyperactivity is implicated in B-cell malignancies and autoimmune diseases. This review derives the structural complementarity of the BTK-kinase domain and its inhibitors from recent three-dimensional structures of inhibitor-bound BTK in the protein data bank (PDB). Additionally, this review analyzes BTK-mediated effector responses of B-cell development and antibody production. Covalent inhibitors contain an α, β-unsaturated carbonyl moiety that forms a covalent bond with Cys481, stabilizing αC-helix in inactive-out conformation which inhibits Tyr551 autophosphorylation. Asn484, located two carbons far from Cys481, influences the stability of the BTK-transition complex. Non-covalent inhibitors engage the BTK-kinase domain through an induced-fit mechanism independent of Cys481 interaction and bind to Tyr551 in the activation kink resulting in H3 cleft, determining BTK selectivity. Covalent and non-covalent binding to the kinase domain of BTK shall induce conformational changes in other domains; therefore, investigating the whole-length BTK conformation is necessary to comprehend BTK’s autophosphorylation inhibition. Knowledge about the structural complementarity of BTK and its inhibitors supports the optimization of existing drugs and the discovery of drugs for implication in B-cell malignancies and autoimmune diseases.

## 1. Introduction

The complementary molecular framework of cells is essential and contributes to biological mechanisms. Comprehension of the physiological processes necessitates a prior understanding of complementarity—i.e., a physiological process depends on two biological components that are mutually exclusive [1]. Structural complementarity between biological macromolecules and their small molecule messengers is the impetus for molecular recognition and efficient binding, which guides several physiological mechanisms [2]. The agonist or antagonistic interaction of a small ligand or drug with a receptor exhibits a succinct lock-and-key-like structural complementarity [3]. The structural complementarity of biomolecules in shape, size, and functional groups (bonding and charge complementarity) at the reaction interface drives the intended pharmacological process [2,4]. 

Bruton’s tyrosine kinase (BTK) is a member of the TEC family of non-receptor tyrosine kinases and plays a vital role in several receptor-mediated signaling pathways. BTK is a crucial component in the signal transduction pathway of a transmembrane receptor, B-cell receptor (BCR), in B lymphocytes. B-cell survival and proliferation are primarily mediated through antigen-induced BCR signaling, initiating the activity of a series of protein kinases like LYN, Spleen tyrosine kinase (SYK), BTK, and highly expressed isoform p110 of PI3K, and the downstream signaling cascade mediated through Phosphoinositide-3-kinase, AKT (Protein kinase B) and mitogen-activated protein kinases (MAPK) [5]. Likewise, BTK also aids the survival and growth of malignant B-cells [6]. BTK expression is evident in hematogenic cells and innate immune cells like macrophages, monocytes, mast cells, and basophils [7]. BTK activation plays a significant role in signaling downstream mediators of autoimmunity and inflammation. COVID-19 patients with severe lung inflammation showed aberrant BTK activity. BTK inhibitors were repurposed to tackle COVID-19 due to their modulatory effect on autoimmunity and hyperinflammation [8]. Therefore, grasping the structural data about the function of BTK is vital for designing inhibitors targeting BTK for B-cell malignancy, autoimmune diseases, and COVID-19 therapy. Numerous reviews on BTK’s role in B-cell cancers and the clinical development of covalent inhibitors of BTK are available. However, literature focusing on molecular-level interactions of the inhibitors with BTK and alterations of BTK conformation in response to inhibitor binding is scarce. This review aims to explore the conformational changes of BTK and its intermolecular interactions with inhibitor drugs to reveal the structural complementarity, in addition to outlining the signal transduction and BTK’s role in B-cell malignancies and autoimmune diseases. The review also is a critical insight into the differences in molecular interaction mechanisms of covalent and non-covalent inhibitors of BTK. It is important to address the structural requirements for BTK’s covalent and non-covalent inhibitors for efficient binding, improved potency, and fewer side effects. This comprehensive review will guide and foster the successful discovery of BTK inhibitors. The review consists of eight sections; Section 2, Section 3, Section 4, Section 5, Section 6 and Section 7 which brief the role of BTK in BCR signaling, its association with different malignancies and autoimmune diseases, and the approval status and chemistry of covalent and non-covalent inhibitors. For obtaining data related to the above sections, the literature search included keywords BTK, BTK in BCR signaling, BTK covalent inhibitors, and BTK non-covalent inhibitors. Appropriate information was retrieved to represent the advances in the field and to describe basic pharmacology. For the preparation of Section 8, we examined the 3D structures of BTK with inhibitor ligands deposited in the Protein Data Bank (PDB) from 2020 to the present date to deduce their complementary binding structural characteristics.

## 2. Structure of Bruton’s Tyrosine Kinase and BTK in BCR Signaling

BTK’s structural domain is composed of five different protein interaction sites: an N-terminal (-NH_2_ terminal) pleckstrin homology (PH) and TEC homology domain (TH), a proline-rich region (PRR), two SRC homology domains (SH3 followed by SH2), and a C-terminal (-COOH terminal) kinase domain (KD) with enzymatic activity [9]. These domains bind to different cytosolic proteins and transcriptional factors to mediate the cell signaling pathways. PH-TH and PRR domains contain a zinc-finger motif that is important for the optimal activity and stability of the protein [10]. BTK presents an inactive conformation in the cytoplasm in which the SH2 and SH3 domains stabilize the inactive conformation of the kinase and are only transiently recruited to the membrane upon activation [11]. 

BTK activation requires two crucial steps. First, upon binding of specific cytokines to its receptor or antigens to the cells, BTK is phosphorylated at position Tyr551 in the kinase domain by SYK or SRC family kinases. Phosphorylation of BTK occurs by interaction of the binding site of the PH-TH domain and phosphatidylinositol-3,4,5-triphosphate (PIP3) membrane lipid. Phosphorylation of BTK at the Tyr551 in the kinase domain results in autophosphorylation at position Tyr223 in the SH3, stabilizing the active conformation and fully activating BTK kinase activity [11]. Figure 1 is the 2D representation of domain arrangement in the whole BTK. Experimental reports or protein data bank depositions on the 3D structure of BTK containing all its domains are unavailable. A PDB structure of BTK, 4XI2, with a maximum of three domains and corresponding linkers, is used for representing the whole structure of BTK. Figure 2 shows the 3D structure of the whole BTK. 

The kinase domain of BTK is the target for covalent and non-covalent inhibitors in the therapeutic intervention of B-cell malignancies [12]. Kinase domain has two nodes: -NH_2_ terminal and -COOH terminal nodes. The active site is located amidst these nodes. The -NH_2_ terminal node comprises five β-sheets running anti-parallelly and two α-helices. The -COOH terminal node comprises seven α-helices and six β-sheets. The ATP-binding cleft is situated between the two nodes. The two nodes have complementary rotations and conformations, exposing the ATP binding cleft when in the active state and vice versa in the inactive state [13]. The catalytic site residue Tyr551 is located in the active loop of the -COOH node. The two nodes are linked by a short flexible polypeptide fragment that enables their complementary movements. 

The BCR signaling is mediated by multiple cytokine receptors, G-protein coupled receptors, and antigens. Moreover, the activation of BTK is regulated by several tyrosine kinases, such as JAK, SYK, LYN, and FAK family kinases [5]. Antigen binding that initiates immune response requires the presence of membrane-bound immunoglobulins on the surface of the B-cell. Antigen-bound immunoglobulin undergoes conformational change leading to the activation of LYN, progressing to the ITAM phosphorylation, thereby generating the SYK binding site [14]. Activation of LYN initiates recruitment of signalosome, a complex of tyrosine kinases, and a range of LYN- and SYK-binding proteins, BTK, BLNK, SHC, Grb2, and BCAP(PI3K) [15]. SYK phosphorylates BLNK linked to CD79a, which phosphorylates BTK via BTK’s SH2 domain [16]. BTK and SYK are vital in steering the BCR distal signaling through interaction with BLNK [17]. The involvement of BTK in BCR activation reflects in various stages of B-cell development, including differentiation, maturation, proliferation, and apoptosis [18]. Moreover, phosphorylation of the tyrosine domains of cytoplasmic B-cell co-receptor CD19 by LYN provides a binding site for the adaptor protein-cell phosphatidylinositol- 3-kinase (PI3K). The docking of PI3K, particularly its p110δ isoform, to CD19, assists its interaction with plasma membrane components, thereby contributing to BCR signaling downstream transmission [19]. PI3K phosphorylates PIP2, an essential membrane lipid, to PIP3. PIP3 has the inherent character of interacting with proteins having pleckstrin homology (PH) domain. Hence, BTK is mobilized to the plasma membrane, resulting in PLC-γ2 activation. In addition, active PIP3 recruits AKT to the plasma membrane due to its interaction with AKT’s PH domain. AKT is activated in the plasma membrane by 3-phosphoinositide-dependent protein kinase 1 (PDK-1) and protein complex mammalian target of rapamycin (mTORC2) through phosphorylation at T308 and S473 residues, respectively, leading to utmost BCR activation [20]. 

Hydrolysis of PIP2, mediated by the phosphorylation of PLC-γ2, leads to the generation of inositol-3,4,5-triphosphate (IP3) and diacylglycerol (DAG). Together with IP3 and DAG, calcium activates NFAT and β isoform of protein kinase-C (PKCβ), respectively [21,22]. PKCβ activation results in the activation of mitogen-activated protein kinase (MAPK), Jun N-terminal kinase (JNK), and extracellular signal-regulated kinases 1 and 2 (ERK1 and ERK2) pathways contributing to B-cell growth, survival, and apoptosis [23]. PKCβ, through its effect on constituents of NF-κB signaling and p38, influences cell proliferation and apoptosis [24]. The generation of NFAT and NF-κB through downstream signaling of PLC-γ2 affirms BTK’s involvement in post-BCR ligation [25]. To add on, MAPK can also be triggered by another PLC-γ2 mediated signal, RAS oncoprotein, which is involved in RAS stimulation. RAS generation in the signal transduction revolves around the growth factor receptor-bound protein2 (Grb2) alongside VAV, which escorts SOS protein complex formation. RAS-GTP, an active form of RAS observed after its association with signalosome, leads to activation of RAF kinases and induction of ERK1/2 phosphorylation, resulting in transcription of c-fos and c-jun genes that are vital for cell viability [26]. Figure 3 illustrates the cascade of events showing BTK’s participation in BCR signaling influencing B-cell development, endurance, and proliferation.

## 3. Association of BTK with B-Cell Malignancies

BCR expression is evident in most B-cell lymphoma cells that propagate BCR signaling in the intact B-cell for malignant cell growth [27]. BTK activity in B-cell differentiation, proliferation, and survival also applies to the microenvironment of B-cell lymphomas [28]. Dysregulation of the BCR signal transduction in a normal cell may trigger B-cell malignancies due to alterations in the gene arrangement and translocation of chromosomes [27]. Failure of BTK function leading to BCR malfunction results in poor B-cell tolerance [29]. Activation-induced cytidine deaminase (AID) instigation is essential for improved B-cell tolerance. AID is found in the germinal center of B-cells and regulates mature B-cell apoptosis [29]. AID activation leads to somatic hypermutations of DNA, resulting in alterations in Ig specificity. AID-led DNA mutations and chromosomal translocations are eventually accompanied by the rearrangement of heavy chain immunoglobulin gene (IgH) by proto-oncogenes, namely BCL2 and cYMC that make cancer cells resistant to apoptosis. AID overexpression is also observed in B-cell lymphomas [30]. BTK overexpression increases germinal center generation, leading to dysregulated homeostasis of T-cells, thus explaining the pathological role of BTK in influencing BCR signaling in B-cell lymphomas. The association of BTK with different B-cell malignancies is discussed below.

### 3.1. Chronic Lymphocytic Leukemia (CLL)

The presence of aggregated CD5+ B-mature cells in the systemic circulation characterizes CLL. CLL B-cells are manifested by a decrease in immune response towards BCR ligation, suggesting prolonged BCR localization inside the cell and signaling, along with less expression of IgM on the cell surface [31]. The presence of active BCR cell gene expression in CLL isolated from lymphatic cells indicated the existence of active BCR signaling. The defective IGHV gene with ‘stereotyped BCR receptors’ was also identified in CLL [32]. CLL pathogenesis is initiated by pressure from antigens of cell apoptosis or pathogens and is associated with high BTK activity, leading to an aberrant increase in BCR signal [33,34]. It was surprising to find ibrutinib-treated CLL cells have the efficacy to inhibit CXCL13- and CXCL12- induced migration, which proves BTK signaling in cell migration to lymph nodes’ proliferation centers in CLL [35,36]. Various findings show the significant role of BTK-aided signaling in the commencement and maintenance of CLL. The existence of phosphorylated BTK in a fraction of CLL samples also ascertains the overexpression of BTK in CLL B-cells [37]. Tumor cessation in mouse models that lack BTK complement these observations [18,38].

### 3.2. Mantle Cell Lymphoma (MCL)

MCL is a lymphoproliferative disease confining to post-germinal center in origin, and has its mantle zone differentiated by the presence of malignant B-cells. A remarkable feature alongside CLL was the appearance of a restricted BCR gene repertoire, which paved the way to assume the role of BCR signaling in the progression of MCL [39]. Overexpression and activation of BTK by Tyr223-phosphorylation also led to the onset of SYK stimulation, a resultant effect of BCR cross-linking. These effects expedited the survival of malignant cells by releasing autocrine factors and adherence to human bone marrow stromal cells, which also correlated well with the deregulation of the kinases in the BCR signaling [40].

### 3.3. Diffuse Large B-Cell Lymphoma (DLBCL)

The presence of fast-growing malignant B-cells in the nodal and extranodal sites with single or multiple focal points identifies DLBCL. This B-cell non-Hodgkin’s lymphoma has been categorized as activated-B-cell-like (ABC-DLBCL), GC B-cell-like (GCB-DLBCL), and primary mediastinal B-cell lymphoma (PMBL), based on gene expression profiling. Investigations reveal that the former two lymphomas weigh more in clinical cases, with the latter to be accounted for the least. BCR signaling in ABC- DLBCL highly depends on the NF-κB pathway-mediated anti-apoptotic events that are conciliated by CARMA1 mutations and loss of function mutations of modulators turning down NF-κB [41]. Findings suggest that there is BCR amplification and protracted triggering of AKT due to the enhancement of BCR expression in ABC-DLBCL [42]. Survival of DLBCL cells largely relies on upregulated BCR signaling [17]. Amplification of malignant cells is mediated through BCR downstream signaling activation involving NF-κB and PI3K pathways. The association of DLBCL with components of BCR channels can be either antigen-dependent or via mutation of downstream components in the pathway, like CARMA1 and NF-κB negative regulators [5]. Gene expression in ABC-DLBCL was found to be influenced by IgM+. The impact of antigens in pathogenesis points to the existence of stereotyped BCR receptors, and the role of autoantigens can be matched with various BCR specificities. This auto-antigenic mediated response to BCR in ABC-DLBCL ascertains the BCR activation hinged with antigen [35].

### 3.4. Burkitt’s Lymphoma (BL)

BL is an aggressive malignancy caused by unregulated expression of MYC due to abnormal translocation of the MYC gene into the loci of immunoglobulin on the cell, resulting in the selection of the cell for expressing BCR. Blocking of PI3K-generated post-BCR ligation on BL, either directly or selecting its substrate during the activated state as mTORC1, can lead to cell death. When these facts are combined, it is evident that Burkitt’s lymphoma phenotype resulted from an activated PI3K and MYC combination [43]. BL cells were found to be receptive towards SYK expression and depletion of CD79a, contrary to its resistive behavior observed with the knockdown of BTK and CARMA1 [44,45]. The mutated form of transcription factor TCF3 in BL leads to elevated expression of BCR mediated through PI3K stimulation [26].

## 4. Association of BTK with Other Cancers

In addition to its dominating space in signal transduction mediated through BCRs, increasing evidence shows the impact of BTK in other subtypes of cancers. B-cell and macrophage-mediated T-cell suppression conciliate the role of BTK in pancreatic adenocarcinomas. Association of BTK was evident in a study using the BTK inhibitor ibrutinib on pancreatic ductal adenocarcinoma (PDAC)-bearing mice, which resulted in the termination of PDAC growth by deviating macrophages to target T(H)1 phenotype involved in CD8(+) T-cell cytotoxicity paralleled with a boost in chemotherapy responsiveness [46]. Studies unveiled that CD1dhiCD5+ regulatory B-cells (Breg) differentiation in the pancreatic tumor was regulated by BTK activation through CD40 and IL-6 receptors aside from Myd88 and IRAK-1, effectors in the downstream signaling of cytokine molecule IL-1 that leads to NFκB-mediated pro- and anti-inflammatory cytokine expression [47,48]. BTK inhibition results in the inhibition of differentiation of Breg and secretion of IL-10 and IL-35, which mediates Breg’s immunosuppressive action [48]. Studies suggest an upregulation of BTK protein with activated mTOR signaling in CD133+-SP Bladder cancer (BLCA) cells [49]. BTK inhibition is crucial to inhibit breast cancer metastasis. BTK regulates breast cancer metastasis by activating PLC γ2/PKC signaling. Activated PKC mediates MAPK and NFκB signaling. The TPA, 12-O-tetradecanoylphorbol-13-acetate is pivotal for generating and releasing matrix metalloproteinases (MMP-9) through activation of PKC, and MAPK controls TPA’s action. MAPK also regulates activator protein-1 (AP-1) and NFκB, strongly influencing TPA-led MMP-9 activation. MMP-9 promotes the cell invasion property of breast cancer cells. Hence, BTK inhibition shall inhibit the expression of MMP-9 initiated by TPA, thereby suppressing breast cancer metastasis [14].

## 5. Association of BTK with Autoimmune Diseases

BCR signaling also activates BTK in cells like mast cells, basophils, monocytes, macrophages, and osteoblasts involved in hematogenesis and autoimmunity [7]. FCγ and FCε receptors in macrophages and mast cells are modulated by BTK signaling, shown in Figure 3. Antigen binding to these receptors activates LYN and SYK, which activates BTK, resulting in the activation of the MAPK pathway and leading to the release of inflammatory cytokines [50]. Toll-like receptors (TLR), on recognizing disease-specific, damage, and pathogen-associated structural patterns (DAMP and PAMP), activate host immunity. BTK is essential for the TLR-mediated release of inflammatory cytokines [51]. Thus, the depletion of BTK activity shall be helpful in the prognosis of autoimmune diseases.

## 6. BTK Inhibitors

Having discussed the role of BTK in BCR signaling and the development of B-cell malignancies, it is implied that BTK inhibition is vital for treating B-cell malignancies. Knowledge of BTK domain arrangements and their complementarity for small molecule binding has steered the design and development of BTK inhibitors (BTKi). Two classes of BTKi are available: covalent inhibitors and non-covalent inhibitors. Covalent inhibitors of BTK (BTK-Covi), chemically, are Michael acceptors establishing a covalent bond with catalytic Cys481 in the ATP binding site of the kinase domain. BTK-Covi are known as irreversible BTK inhibitors. Non-covalent inhibitors (BTK-Ncovi) occupy the kinase domain of BTK, establishing weak, reversible inter-molecular interactions like hydrophobic and electrostatic interactions with its residues, and are also known as reversible BTK inhibitors [52].

### 6.1. Covalent Inhibitors of BTK

The binding of BTK-Covi with the active site of BTK blocks the binding of ATP, thereby hindering the BTK autophosphorylation and inhibiting BTK. Inhibition of BTK also prevents the phosphorylation of the kinases acting downstream in the B-cell signal transduction [53]. Ibrutinib, the prototype BTK-Covi, suffers from side effects because of non-specific blockade of other targets like EGFR, ErbB2, ITK, and TEC, as these proteins harbor a conserved cysteine residue that aligns with Cys481of BTK [6,54,55,56]. Table 1 has information on a few of the approved BTK-Covi. Concern about the side effects of ibrutinib set the investigation for more selective BTK inhibitors with minimal toxicity. Acalabrutinib and zanubrutinib, comprising the second generation of BTK inhibitors, offer more selective BTK inhibition. The inhibitory effects on TEC were less with an absence of ITK or EGFR blockade that was regarded as their advantage, although these inhibitors’ binding occurred at Cys481. Acalabrutinib lacks adverse effects like severe diarrhea and rashes that were assumed to be linked with EGFR inhibition, and platelet dysfunction associated with blocking TEC [57,58,59]. Compared to ibrutinib, zanubrutinib owes good pharmacokinetic data and the absence of the inhibition of non-BTK targets (ITK or EGFR) [58].

### 6.2. Resistance to Covalent Inhibitors of BTK

Like other anti-cancer drugs, adverse effects and resistance to the covalent BTK inhibitors were observed. Nearly one-third of the patients treated with ibrutinib developed primary resistance, and a few others have shown secondary (acquired) resistance [60]. Acquired resistance to ibrutinib was seen in multiple B-cell lymphomas. Among the CLL patients treated with ibrutinib, mutations in the Cys481Ser (ibrutinib-binding), Thr474Ile, Met, Ser (gatekeeper), and Thr316Ser (SH) were observed, positions shown in Figure 2 [61]. 

The Waldenström’s macroglobulinemia (WM) patients on ibrutinib therapy commonly showed Cys481 mutation. Due to these mutations, there is an interference in the binding of the drug to the protein, leading to the development of resistance [62,63]. Other types of mutations that were observed in the WM patients were the PLCY2 and CARD 11. Mutations in the IGHV in patients suffering from WM and mutations in the MYD88 in patients with diffuse large B-cell lymphoma were also associated with the development of primary resistance [61]. 

According to a study that evaluated patients with relapsed CLL, it was found that patients who were treated with ibrutinib experienced a relapse due to resistance. Whole exome sequencing (WES) performed on the blood samples of patients revealed a mutation of cysteine to serine at position 481. These alterations of the Cys481 residue impaired the binding of the inhibitor drug to BTK. Another type of mutation observed in a few patients was due to the change of arginine to tryptophan in PLCY2 at position 665. One patient showed both BTK and PLCY2 mutations. However, the resistance observed was mainly due to mutations in Cys481 and less commonly due to mutations in the PLCY2 [61,64]. CLL patients treated with other covalent BTK inhibitors also showed disease progression due to the development of resistance [65,66].

WES revealed Cys481 mutation as the primary mechanism for developing ibrutinib resistance in the relapse of MCL tumors [67,68]. It was revealed that one-third of the MCL patients are ibrutinib-resistant, and the other patients who were sensitive to the drug developed resistance [69]. MCL patients experienced primary and secondary resistance to ibrutinib [70]. Multiple resistance mechanisms in MCL include recurrent mutation of Cys481Ser or PLCY2 [71].

## 7. Non-Covalent Inhibitors of BTK

Resistance to BTK-Covi and the adverse effects of BTK-Covi led to the development of non-covalent inhibitors. BTK-Ncovi does not bind to Cys481 of BTK and helps overcome ibrutinib resistance due to mutation in Cys481. Given the role of BTK in autoimmunity, BTK-Ncovi are under preclinical trials against autoimmune disorders. Vercabrutinib and fenebrutininb are typical examples of BTK-Ncovi which are under clinical trials. These drugs are reported to possess fewer off-target adverse effects [72]. None of the BTK-Ncovi are approved for clinical use. Table 2 provides information about BTK-Ncovi under clinical trials [73]. Recently, resistance to BTK-Ncovi has been reported. Resistant mutations occurred at residues in the BTK kinase domain Val416Leu, Ala428Asp, Met437Arg, Thr474Ile, Leu528Trp, and on the BCR signal mediator PLC-γ2 [74].

## 8. Structural Complementarity of BTK and Its Inhibitors

We examined the 3D structures of BTK with its inhibitors recently deposited in the protein data bank (PDB) to update the knowledge about structural attributes of small inhibitory ligands complementary to BTK inhibition. 

The PDB search was directed towards the ligand-bound kinase domain of BTK wild-type structures obtained from *Homo sapiens*. Structures that the same research groups deposited were examined for the presence of the most active compound reported, and only such structures were used for the analysis. Structures of the kinase domain were only considered for the study. Thirty X-ray crystal-derived structures of BTK bound to small molecule inhibitors were retrieved. Thirteen structures were removed from the study because of similarity in structure due to the presence of similar scaffolds in ligands reported from the same research group. Eight structures were omitted because the ligands interacted with the PH domain of BTK mutants. One structure was not utilized as it was BTK apo form. Thus, eight recently deposited PDB structures of *Homo sapiens* BTK-kinase domain-bound inhibitors made up the final analysis stage. Table 3 contains the information on 3D structures analyzed for structural characteristics of ligands for binding to BTK.

### 8.1. Structural Complementarity of Covalent Binding to BTK

Though no new BTK structures bound to ibrutinib were retrieved from PDB, assessing the ibrutinib binding reported in recent studies is inevitable. As a prototype molecule, ibrutinib, a covalent irreversible BTK inhibitor, has undergone rigorous scrutiny to unveil possible inhibitory binding mechanisms [75]. Ibrutinib binds to the ATP binding site located in the kinase domain. Ibrutinib establishes a covalent bond with the Cys481 residue of the ATP binding site. The BTK kinase domain undergoes conformational changes on the covalent binding of inhibitors. Conformations of αC-helix and activation kink in the BTK kinase domain are important indicators to conclude the stabilized apo or ligand-bound conformations. The αC-helix equilibrates between inactive (αC-helix-out) or active (αC-helix-in) spatial arrangements that are controlled by a structural plug in the helix, which regulates the existence of a salt bridge between glutamic acid-lysine residues in the kinase domain [76]. It has been proved that Trp395 (W395), in Figure 2, undergoes a conformational change in response to the spatial orientation of the αC-helix. A recent study that utilized NMR and mass spectrometry to track the spatial orientations of the kinase domain and the whole BTK structure on ibrutinib binding concluded that αC-helix was stabilized in an inactive-out conformation disrupting the catalytic activity of BTK. Figure 4 illustrates the stabilized conformation of the kinase domain of BTK bound to ibrutinib. In addition, ibrutinib induced a change in the conformation of Trp251 (W251) in the SH3 domain, altering its conformational preference far from the autoinhibitory BTK conformation. Alteration in SH3 conformation indicates that ibrutinib binding causes stereochemical changes in the whole structure of BTK apart from the catalytic kinase domain. Comparing the covalent binding of ibrutinib with other covalent inhibitors like dasatinib and CC292 indicated that the final stabilized conformation of the kinase domain of BTK was different with different inhibitors [77]. Although ibrutinib and dasatinib were found to bind to Cys481 through a covalent bond, the activation kink conformation was different. The dasatinib-bound BTK kinase domain did not display the activation kink close to the αC-helix, whereas the he ibrutinib bound-BTK kinase domain did display the activation kink close to the helix. Despite stabilizing αC-helix-in state, dasatinib is capable of BTK inhibition because it does not affect the regulatory domain orientations; hence, BTK is in its autoinhibitory inactive conformation. Ibrutinib and dasatinib occupied the posterior cleft, while CC292 occupied the anterior cleft in the catalytic binding site of the kinase domain. Ibrutinib and dasatinib existed in an extended conformation inside the binding site, while CC292 acquired a U-shape during binding. Figure 5 displays the overlay of the BTK kinase domain bound to ibrutinib, dasatinib, and CC292. CC292 did not produce any conformational change in the whole length of BTK, except the kinase domain, while ibrutinib and dasatinib did. All of the above observations indicate that there exists a structural complementarity between inhibitors and BTK, resulting in characteristic selectivity and adverse effect profiles. 

To further support the structural complementarity of BTK and its covalent inhibitors in terms of the functionalization of chemical moieties, we consider the results of a mechanistic study reported. The mechanistic analysis of the low energy reaction between Cys481 of the kinase domain of BTK and ibrutinib confirms that the reaction mechanism is a nucleophilic addition of S-alkyl of CYs481 to the double-bonded carbons in ibrutinib. Negatively ionized Cys481 attacks positively charged ibrutinib that results in an enol ibrutinib-BTK complex which then undergoes tautomerization to produce the stable keto form of ibrutinib-BTK complex. An α, β-unsaturated ketone (or aldehyde moiety) functioning as the chemical fuse is essential for initiating a covalent Michael addition reaction. First, Cys481 transfers H^+^ to the keto oxygen of ibrutinib, producing an ionic pair of S-Cys and HO^+^ibrutinib, followed by the nucleophilic attack of this ionized ibrutinib by Cys481. The mechanistic study also indicated that the presence of Asn484 adjacent (maximum 2 residues apart) to Cys481 is essential to stabilize the intermediates of the nucleophilic reaction. It is suggested that Asn484 interacts with Cys481 throughout the reaction, without which the reaction intermediates are not stable [78]. Based on the structures of the available covalent inhibitors and the proposed mechanism of their interaction with Cys481, the pharmacophoric features that complement the binding of covalent inhibitors of the kinase domain of BTK are provided in Figure 6A. Lipophilic rings are required to increase the bulk that facilitates binding site occupancy and establishes hydrophobic interactions with other residues in the catalytic site. 

The PDB structures 7N5Y, 7R60, and 6TFP were analyzed for ligand binding interactions and protein conformation to check for updates on the structural complementarity of covalent ligand binding to Cys481 of the kinase domain of BTK. Overlay of protein structures for conformational analysis and interaction analysis were performed using Chimera and Discovery Studio, respectively. Chemical structures of the novel BTK-Covi fall into the classic model of essential structural features in Figure 6A. The two-dimensional structures of these novel BTK-Covi are provided in Figure 6B. These novel BTK inhibitors have acryloyl amide as the chemical fuse or a Michael acceptor that initiates the nucleophilic reaction with Cys481, resulting in a covalent bond [79,80,81]. TAK020 oriented itself exclusively into the anterior cleft of the binding site in a ‘U’-form, while LOU064 and 2IE were bound to the anterior and posterior clefts in an extended conformation. LOU064 was shown to bind to the inactive BTK conformation and stabilize it in αC-helix-out conformation [79]. Superimposing the ligand-bound BTKs onto ibrutinib-bound BTK suggested that the novel ligands induced similar conformational changes in the BTK kinase domain, like ibrutinib, as shown in Figure 7. Matching the αC-helix conformation of all examined BTK structures shows that these compounds stabilize BTK in its inactive αC-helix-out conformation and also display the activation kink. The orientation of Asn484 in all BTK structures was also indistinguishable from that of ibrutinib-binding, implying that Asn484 shall play a crucial role in stabilizing the interaction of ligands with Cys481. 

The three recently reported covalent inhibitors discussed here exhibited improved potency and reduced off-target effects compared to ibrutinib. Modifying the chemical fuse moiety was impossible as changes led to decreased potency. The heterocyclic spacer can be modified, as seen in remibrutinib; it is aromatic pyrimidine, while in the other two compounds, it is pyrrolidine and piperidine (Figure 6B). Variations in core moieties are also possible, and it varies from bicyclic isoquinoline to monocyclic pyridine or a simple phenyl ring. The terminal hydrophobic rings may be heterocyclic triazole or alkyl/fluoro substituted phenyl rings, which can make the compounds selective to BTK [79,80,81]. 

Together, a structural complementarity exists between the kinase domain of BTK and its covalent inhibitors. Binding site complementarity that involves charges of Cys481 and Asn484 is indispensable when a structurally complementary inhibitor possessing an α, β-unsaturated carbonyl moiety approaches the binding site. The inhibitor chemistry must be complementary to BTK in shape and size and is accomplished by the bulky heterocyclic core adjoining lipophilic rings and a saturated heterocyclic spacer [79,80,81].

### 8.2. Structural Complementarity of Non-Covalent Binding to BTK

Before analyzing the recent depositions in PDB, it is essential to revise the existing knowledge about non-covalent binding to BTK. Non-covalent binding to BTK can be explained by taking fenebrutinib as an example. BTK-Ncovi binds to the kinase domain, and their interactions are not restricted to Cys481 [82]. A detailed structure-activity relationship investigation on BTK-Ncovi revealed the complementary functional groups for enhanced potency, selectivity, and favorable metabolic and safety profile. In addition, the binding site engagement by fenebrutinib was charted out. Fenebrutinib resulted in an induced fit to the kinase domain producing a conformational change in the activation kink that stabilizes BTK in its inactive conformation, shown in Figure 8. The lipophilic tricyclic ring exhibits shape complementarity to the H3 cleft and interacts with Tyr551 in the activation kink, thus inhibiting the autophosphorylation. Fenebrutinib possesses hydrophobic groups filling the H2 cleft and H1 hinge region in an extended ‘U’ conformation. The terminal oxetane-linked piperazine in fenebrutinib is prone to solvent exposure and engaged H2 cleft. The core pyridone ring engaged the H1 hinge [83]. 

We focused on recent BTK-Ncovi optimized for better potency, selectivity, and metabolic stability than the current clinical candidates. Therefore, the inhibitor effect on BTK mutations was not considered. Development in non-covalent inhibitors’ chemistry and BTK binding was ascertained from PDB structures deposited recently. PDB IDs 7KXQ, 6W07, 7LTZ, 6XE4, and 6X3P were investigated for the structural complementarity between BTK-NCovis [84,85,86,87,88]. Overlay of the fenebrutinib-bound BTK-kinase domain on BTKbound to the recent non-covalent inhibitors in Figure 9, shows that the currently developed BTK-Ncovi occupy and interact with the kinase domain of BTK. The catalytic site engagement is also similar to that of fenebrutinib. All novel BTK-NCovi filled the kinase domain’s H3, H2 clefts, and the H1 hinge area. Their primary interaction was with Tyr551 in the activation kink, altering its conformation. Almost all BTK-Ncovi interact with BTK’s inactive autoinhibited conformation stabilizing the αC-helix-out conformation, resulting in extended docking of ligands leading to prolonged inhibition [83,84]. 

Research on BTK-Ncovi is progressing toward optimizing selectivity, potency, metabolic stability, and reducing side effects [52,89]. Non-covalent inhibitors with the complementary structural features provided in Figure 10A portray excellent BTK inhibition and fewer side effects [84,90]. In-vitro and in-vivo side effects of non-covalent inhibitors decreased when functional groups engaging H3 and H2 clefts were optimized, such that H3 binders are highly lipophilic bicyclic/tricyclic planar rings and H2 binders are less lipophilic medium-size rings [91]. The H3 cleft of BTK is versatile in accommodating binders like bulky, lipophilic t-butyl-oxadiazole, t-butyl-triazole, isopropyl-oxo-azetanyl, trifluoromethyl pyridyl, and t-butyl-pthalazinyl moieties (Figure 10B). These moieties also maintained hydrophobic or polar interactions with Tyr551, increasing the potency [83,90]. Moreover, H3 binding groups afford selectivity to BTK, thereby reducing the off-target effects [84]. H2 cleft is a hydrophobic flat binding region, and its interaction with specific chemical moieties may also provide selectivity. It has been proved that the H2 cleft can hold a maximum of three rings adjoined by single- or double-atom linkages. H2 binders, if less lipophilic than H3 binders, can offer maximum selectivity and metabolic stability [90,91]. Examples of H2 binders in novel inhibitors include isopropyl pyrazole, methyl pyrazole, bicyclic-fused piperidyl pyrazolidine, and fluoro cyclopropyl groups. H1 binders construct the structure’s core, which usually is an aromatic heterocyclic scaffold or alicyclic core that occupies the hinge region. Novel BTK-Ncovi have cycloheptyl benzene, azepinyl benzene, imidazopyrazine, phenyl pyrimidine, and dipyridine as core functional scaffolds. Appropriate H3, H2, and H1 binders shall improve potency by enhancing the duration of residence of inhibitors inside the binding clefts [85,86,87,88,89,91]. Thus, the BTK-kinase domain provides a complementary binding region for induced-fit docking of non-covalent inhibitors with complementary structures that determines the pharmacodynamic and pharmacokinetic profiles of inhibitors.

### 8.3. Comparison of Binding Interactions of Covalent and Non-Covalent Inhibitors of BTK

Representative molecules from BTK-Covi (TAK-020-bound BTK, 7N5Y) and BTK-Ncovi (BIIB068-bound BTK, 6W07) were considered for comparing the binding interactions. TAK-020, a poly heteroaromatic compound, contains isoquinoline and triazole motifs and produces irreversible covalent inhibition of the BTK-kinase domain implicated in rheumatoid arthritis. It also possesses a pyrrolidine ring, the nitrogen of which is linked to the α, β-unsaturated ketone resembling the acryloyl amide group. TAK-020 shows improved efficacy, potency, and selectivity when compared to ibrutinib. Figure 11A illustrates the covalent binding of the chemical fuse, an unsaturated ketone of TAK-020, to sulfur of Cys481 of BTK within a distance of 1.635 Å. TAK-020 adopts a U-conformation, occupying the ATP-binding site in the kinase domain of BTK. Figure 11B displays the complementary shape and binding groups of TAK-020 capable of establishing hydrogen bonds and van der Waal’s interactions with the BTK-kinase domain. Thr474, the gatekeeper, forms a hydrogen bond with the keto oxygen of triazolone, and Met477 interacts through a hydrogen bond with the N-atom of triazole. This suggests that triazole is complementary in size, shape, and binding groups to fill the hydrophobic binding site, in addition to facilitating binding interactions. The aromatic isoquinoline ring is the core nucleus essential for hydrophobicity and places the compound inside the hydrophobic binding site, complementing the positioning of triazole and pyrrolidine. In addition, isoquinoline also establishes hydrophobic ℼ-alkyl bonds with Val416, Leu528, and ℼ-sigma bonds with Leu408. The pyrrolidine ring functions as a spacer group and is critical in positioning the chemical fuse group close to Cys481 for charge complementarity, initiating the hydrogen ion transfer from Cys481 to the keto group of acryloyl moiety in TAK-020. Pyrrolidine also forms alkyl interactions with Val416. TAK-020 was designed to have a smaller size than ibrutinib with the aim of selective binding to BTK. The heterocyclic spacer in ibrutinib is hexahydro pyridine, which is bulkier than pyrrolidine in TAK-020. The significant difference lies in the lipophilic terminal scaffold; TAK-020 has a medium-size triazolone ring, while ibrutinib has a bulky phenoxyphenyl moiety. TAK-020 does not bind to non-BTK targets like Tec and Src kinases, hence having a better safety profile than ibrutinib. Thus, the complementarity in shape, size, functional groups, and charges on functional groups influence drug efficacy, potency, selectivity, and safety. 

The binding of BTK-Ncovi occurs through an induced-fit mechanism; preliminary interaction with Tyr551 or any adjacent residue like Lys430 in the activation kink induces conformational changes generating H3 cleft specific for BTK [84,90]. For example, Figure 11C,D shows the hydrogen bonding and hydrophobic interactions of the non-covalent inhibitor BIIB068 with BTK. BIIB068 is an orally effective, BTK-selective inhibitor. The side chain isopropyl group attached to the azetidine ring forms complementary ℼ-alkyl interaction with Tyr551, the phosphorylation site in the BTK kinase domain. This hydrophobic interaction alters the conformation of BTK, generating an H3 cleft and placing the azetidine ring deep into the pocket. Furthermore, it favors the angular orientations of the aromatic benzyl reverse amide group into the hinge region and the H2 binder, pyrazolyl pyrimidine, submerged into the H2 cleft. Several other favorable ℼ-sigma and ℼ-alkyl interactions with Leu408, Val416, Ala428, Leu528, and Val546 residues stabilize the active conformation of BIIB068. Unique hydrogen bonds with H1, H2, and H3 residues also stabilize BIIB068 active conformation. Hydrogen bonding happened with the crucial Met477 hinge residue. It is further interesting to note the involvement of water molecules in facilitating the hydrogen bond network with Phe413, Gly414, Lys430, and Arg525 within an interatomic distance of 4 Å. The amide and secondary amine linkers play a significant role in hydrogen bonding, along with pyrimidine nitrogen. Consequently, complementary angular conformational orientation and electrostatic and hydrophobic moieties of non-covalent inhibitors are key features driving non-covalent BTK inhibition. 

## 9. Conclusion and Recommendations

Aberrant BTK expression is evident in B-cell malignancies and autoimmune diseases. BTK has a complex structure, and the elucidation of the whole BTK structure needs attention. BTK inhibition is vital to halt B-cell malignancies and autoimmune diseases. Critical insights into the structure and function of the kinase domain of BTK have boosted the design and development of BTK inhibitors targeting its kinase domain. The foremost requisite for a successful inhibitor is its complementary shape, size, functional groups, and charges that initiate efficient binding to critical binding site residues like Cys481 and Tyr551. Covalent inhibitors of BTK, like ibrutinib and acalabrutinib, are approved for use in B-cell malignancies and autoimmune diseases. Approved covalent inhibitors were appealingly repurposed against severe acute respiratory syndrome coronavirus-2 to improve the hyperinflammatory responses associated with COVID-19. Covalent inhibitors of BTK require an α, β-unsaturated carbonyl moiety functioning as the chemical fuse kicking off the nucleophilic attack by BTK Cys481 residue, ultimately forming a covalent bond. The updated scenario of covalent BTK inhibition has revealed the essentiality of Asn484 at a two-carbon distance to Cys481 for stabilizing the transition state with covalent inhibitors. Covalent binding of inhibitors to Cys481 alters the αC-helix conformation and stabilizes it in an αC-out inactive conformation resulting in the inhibition of phosphorylation of Tyr551 in the activation kink. The covalent binding of inhibitors also induced conformational changes in the SH3 domain, thus inhibiting the autophosphorylation of Tyr223 in the SH3 domain. A functional, structural model for a covalent inhibitor of the BTK kinase domain must possess four distinct chemical moieties: α, β-unsaturated carbonyl fuse, saturated heterocyclic spacer, aromatic heterocyclic core, and lipophilic rings that provide binding site complementarity. Moreover, different covalent inhibitors have different structural features and stabilize the kinase domain in different conformations. Therefore, the binding of covalent inhibitors to BTK needs extensive investigation for further understanding of conformational modifications of kinase and other domains. 

Resistance to covalent inhibitors, their adverse effects, and the void in approved BTK inhibitors for autoimmune diseases paved the way for the discovery of non-covalent inhibitors of BTK. Non-covalent inhibitors are tested against B-cell malignancies and autoimmune diseases, and are established to exhibit safe in-vitro and in-vivo pharmacokinetic properties. Non-covalent inhibitors bind to the kinase domain of BTK through an induced-fit mechanism but not essentially through the covalent bond with Cys481. Structural complementarity of non-covalent inhibitors to BTK includes bulky lipophilic rings as H3 binders for kinase selectivity, heterocyclic core for hinge binding, and lipophilic rings as H2 binders exposed to solvent. Binding to Tyr551 in the activation kink is essential to favor the induced fit of non-covalent inhibitors to H3 and H2 clefts of BTK. This binding to Tyr551 shall occur through electrostatic hydrogen bonds or hydrophobic interactions. Novel non-covalent inhibitors exhibited binding to the PH domain and are effective against primary Cys481 and Thr474 mutations of BTK. Structural features of ligands influencing binding to mutant BTK is not yet clear and studies focusing on conformational alterations in the kinase domain of mutant BTK on ligand binding to the PH domain are unavailable. Hence, BTK mutant structures were not included in the review. Investigations are underway for improving the pharmacokinetic profile of non-covalent inhibitors. Studies on the structural complementarity of non-covalent inhibitors of BTK are scarce; hence, focus on the effect of binding of non-covalent inhibitors on conformations of the kinase domain and whole-length BTK is needed. Therefore, detailed studies on covalent and non-covalent inhibitors of the BTK kinase domain revealing the complementary conformational alterations in the whole structure of BTK are required to enhance the successful design and development of clinically effective drugs for implication in B-cell malignancies and autoimmune diseases.

## Figures and Tables

**Figure 1 pharmaceuticals-16-00400-f001:**
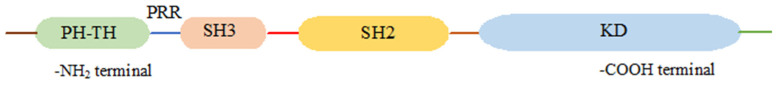
2D representation of domain arrangement in the whole BTK.

**Figure 2 pharmaceuticals-16-00400-f002:**
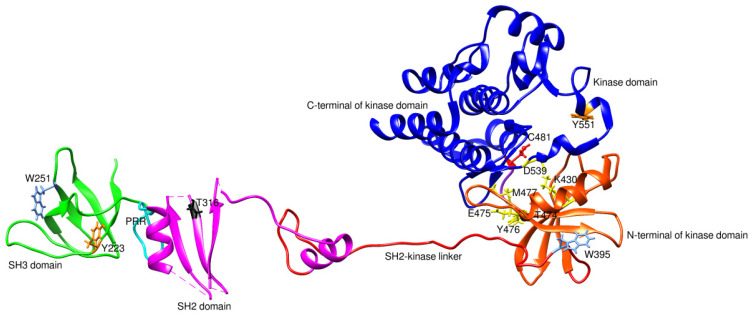
3D structure of the whole BTK (4XI2). PRR is the proline-rich region. Y551 and Y223 are phosphorylation sites. C481 is a covalent inhibitor binding site and the primary site of mutation leading to drug resistance. T316 and T474 are mutation sites conferring drug resistance. K430, E475, Y476, M477, and D539 are residues of the non-covalent inhibitor binding site. W395 and W251 undergo conformational changes in response to conformational changes in the kinase and SH3 domain, respectively.

**Figure 3 pharmaceuticals-16-00400-f003:**
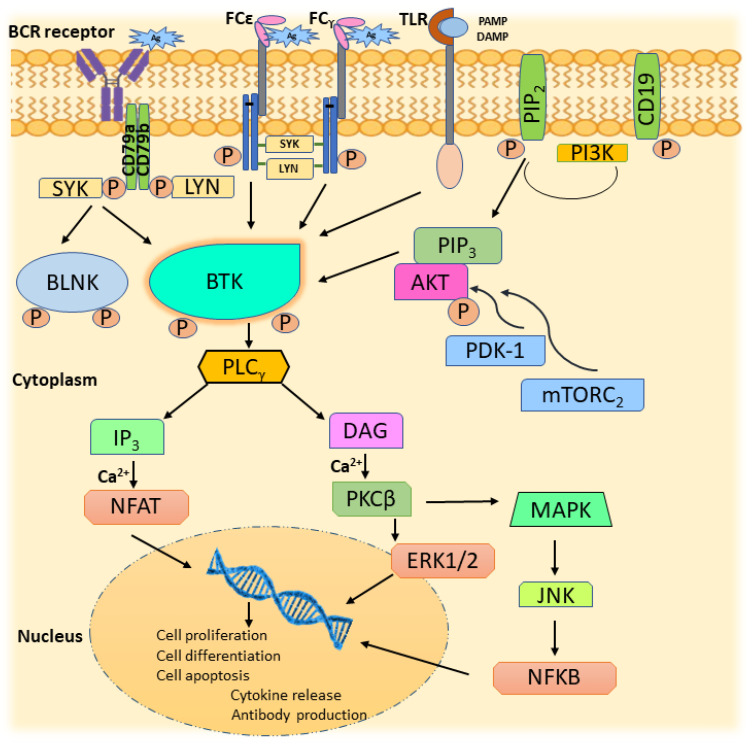
BTK in BCR signaling.

**Figure 4 pharmaceuticals-16-00400-f004:**
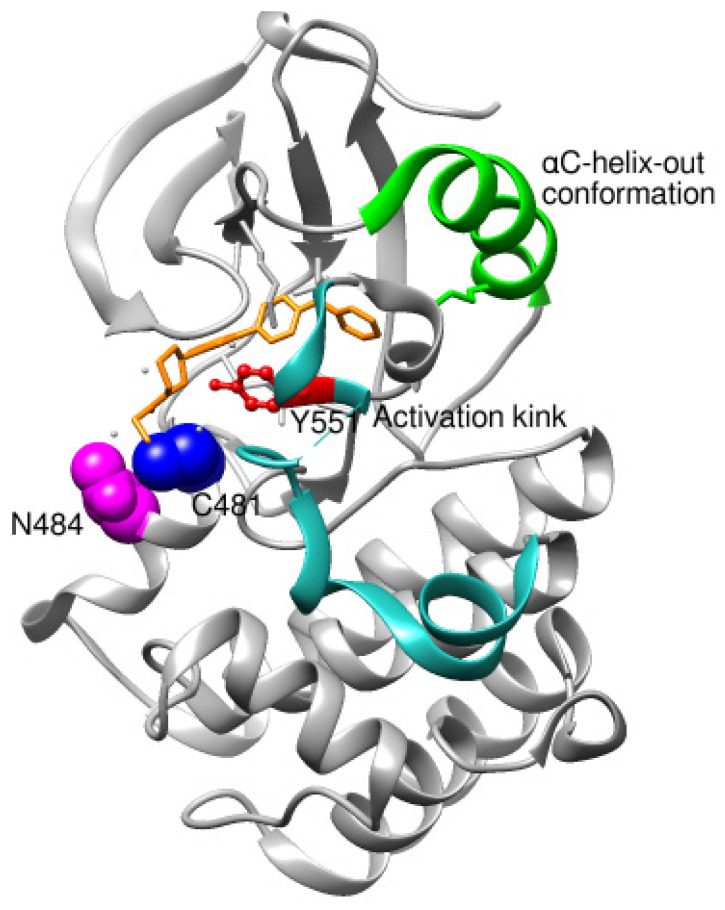
Kinase domain of BTK in αC-helix-out conformation as a result of ibrutinib binding. Asn484 is essential for the stability of intermediates during the nucleophilic reaction between ibrutinib and Cys481.

**Figure 5 pharmaceuticals-16-00400-f005:**
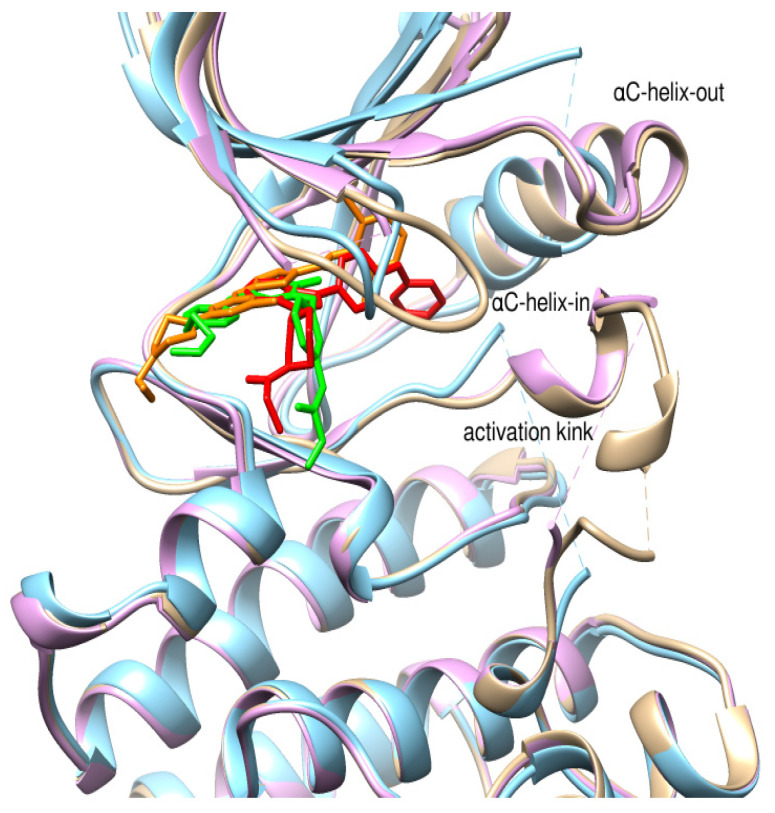
Overlay of BTK kinase domain bound to ibrutinib (5P9J), dasatinib (3K54), and CC92 (5P9L). Protein: grey-ibrutinib-bound, blue-dasatinib-bound, purple-CC292-bound. Ligands: red-ibrutinib, yellow-dasatinib, green-CC292. Ibrutinib and dasatinib interact in an extended conformation and occupy the posterior cleft in the catalytic site. CC292 acquires a U-shape and occupies the anterior cleft in the catalytic site. Dasatinib binding results in an αC-helix-in conformation, while ibrutinib and CC292 stabilize αC-helix-out conformation. Dasatinib-bound BTK did not display the activation kink.

**Figure 6 pharmaceuticals-16-00400-f006:**
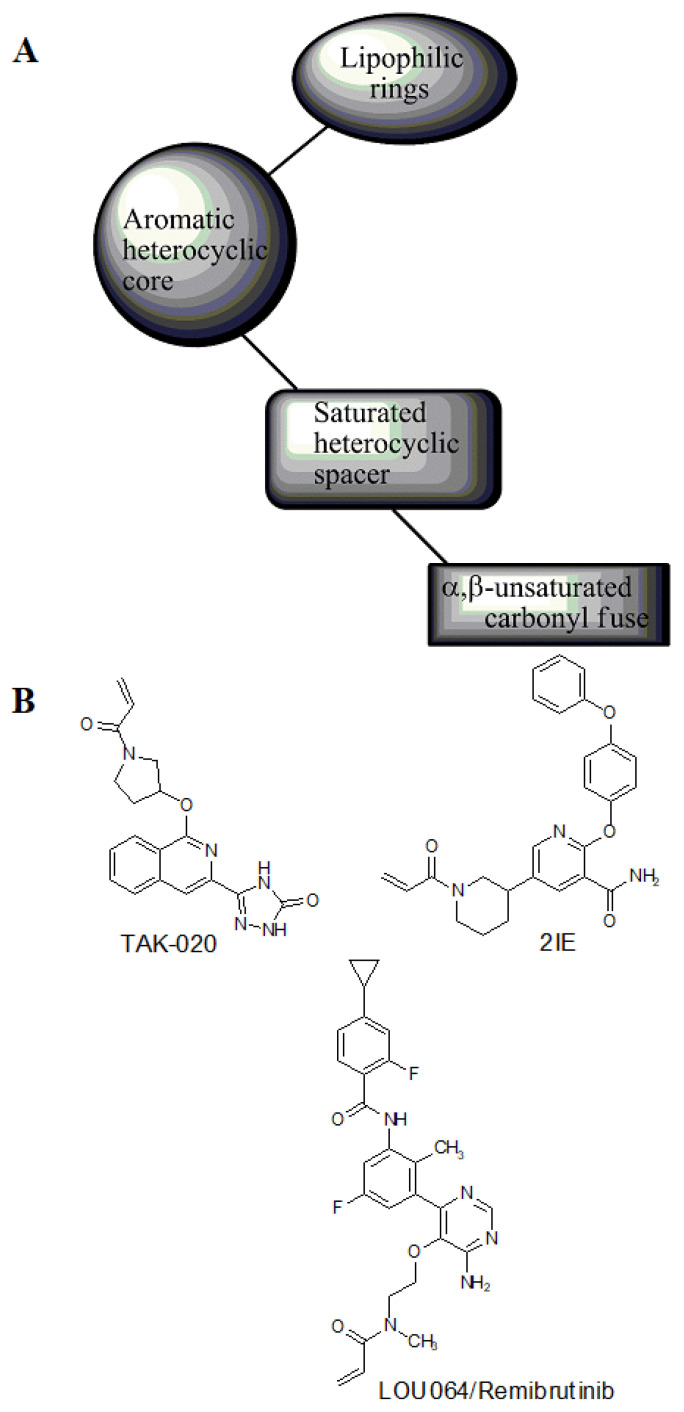
(**A**) Complementary structural features of covalent inhibitors of BTK-kinase domain; (**B**) structures of novel covalent BTK inhibitors deposited in PDB.

**Figure 7 pharmaceuticals-16-00400-f007:**
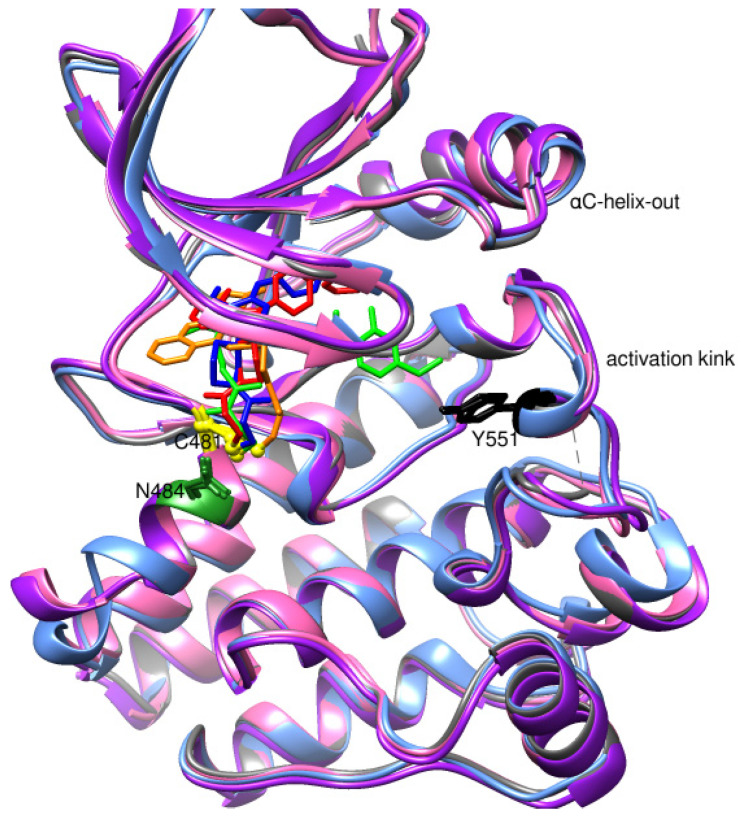
Superimposed 3D structures of novel covalent inhibitors bound to BTK-kinase domain. Protein: grey-ibrutinib-bound (5P9J), blue-LOU064-bound (6TFP), purple-2IE-bound (7R60), pink-TAK020-bound (7N5Y). Ligands: red-ibrutinib, green-LOU064, blue-2IE, orange-TAK020. αC-helix inactive out conformation and activation kink is visible in the other inhibitor-bound BTK conformation, similar to the ibrutinib-bound conformation.

**Figure 8 pharmaceuticals-16-00400-f008:**
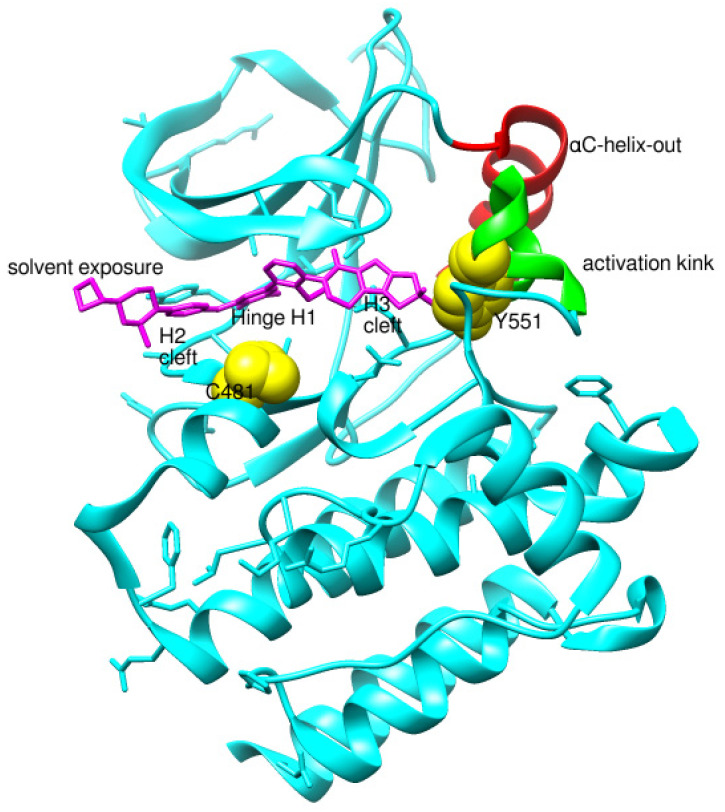
Non-covalent inhibitor fenebrutinib bound to BTK kinase domain (5VFI). Chiefly, the lipophilic moiety occupying the H3 cleft interacts with Tyr551 in the activation kink and inhibits the autophosphorylation of BTK.

**Figure 9 pharmaceuticals-16-00400-f009:**
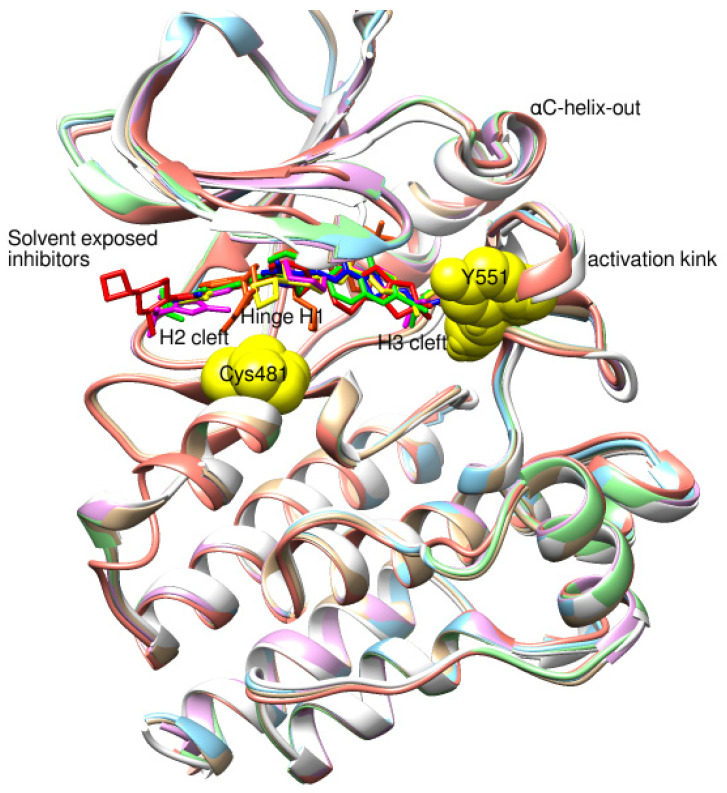
Overlay of 3D structures of novel non-covalent inhibitors bound to BTK-kinase domain. Protein: grey-fenebrutinib-bound (5VFI), blue-X9Y-bound (7KXQ), purple-BIIB068-bound (6W07), cyan-BIIB091-bound (7LTZ), pink-V1G-bound (6XE4), white-L-005298385-bound (6X3P). Ligands: red-fenebrutinib, magenta-X9Y, blue-IIB068, yellow-BIIB091, green-V1G, orange-L-005298385.

**Figure 10 pharmaceuticals-16-00400-f010:**
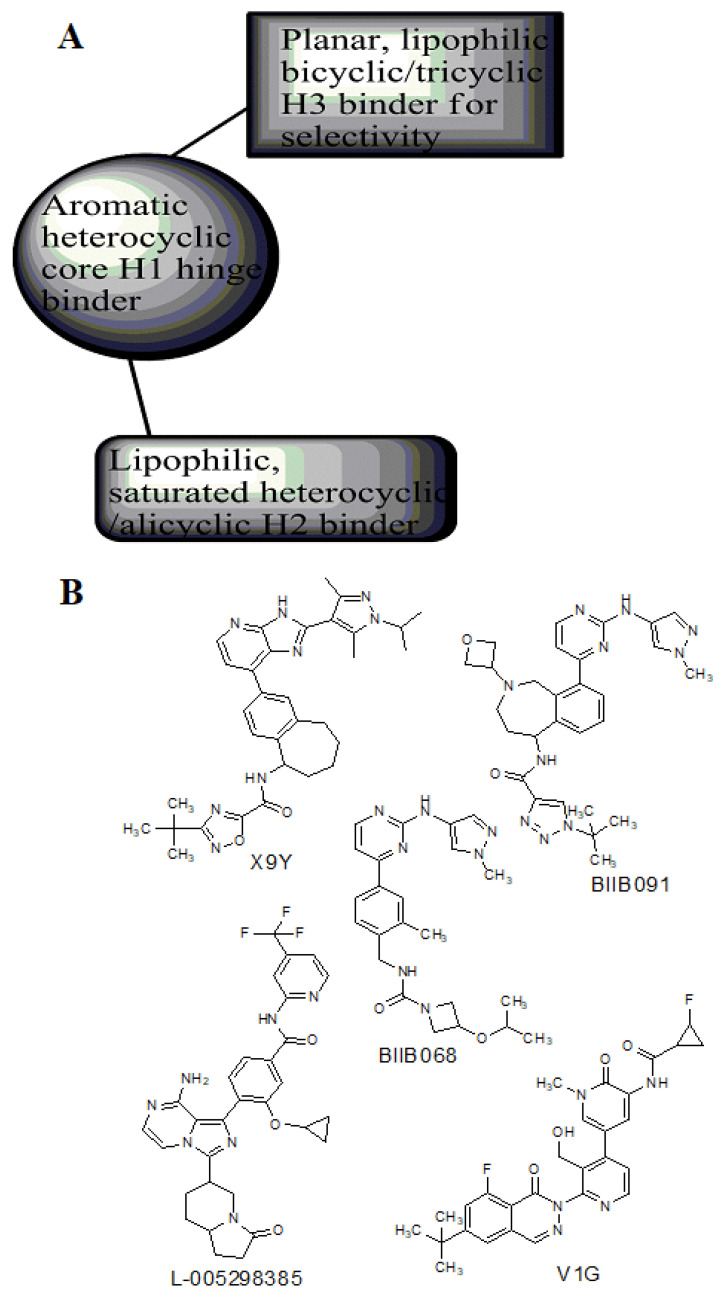
(**A**) Complementary structural features of non-covalent inhibitors of BTK-kinase domain, (**B**) structures of novel non-covalent inhibitors of BTK deposited in PDB.

**Figure 11 pharmaceuticals-16-00400-f011:**
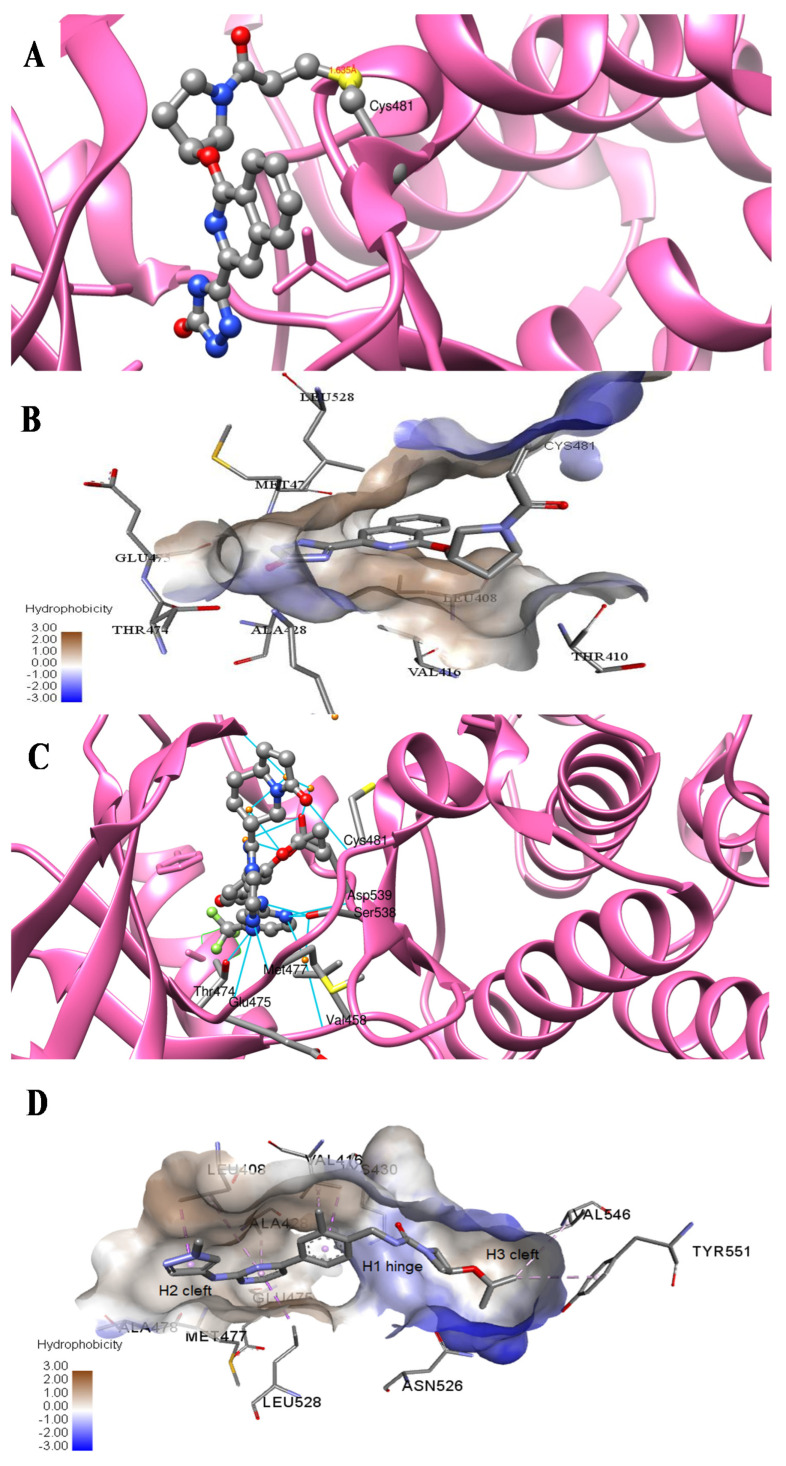
Comparison of binding interactions of covalent and non-covalent inhibitors of BTK-kinase domain. (**A**) TAK-020 interacts through a covalent bond with Cys481 of BTK (7N5Y). (**B**) TAK-020 positioned inside the hydrophobic binding pocket of BTK establishing hydrogen bonds and hydrophobic interactions. (**C**) BIIB068, a non-covalent inhibitor, establishes hydrogen bonds with the BTK kinase domain (6W07). Cys481 is not involved in interaction with BIIB068. (**D**) BIIB068 engagement with Tyr551 creates the H3 hydrophobic cleft for the selectivity of the kinase. Hydrophobic interactions with nearby residues of H1 and H2 sites are also present.

**Table 1 pharmaceuticals-16-00400-t001:** Approved covalent inhibitors of BTK.

Name	Heterocyclic Scaffold	Use
Ibrutinib 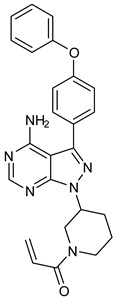	Pyrazolopyrimidine 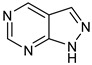	Chronic lymphocytic leukemia, small lymphocytic lymphoma, Waldenström’s macroglobulinemia, marginal zone lymphoma, and relapsed/refractory mantle cell lymphoma
Acalabrutinib 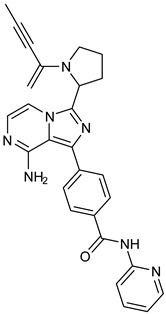	Dihydroimidazopyrazine 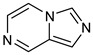	Relapsed/refractory mantle cell lymphoma, chronic lymphocytic leukemia, and small lymphocytic lymphoma
Zanubrutinib 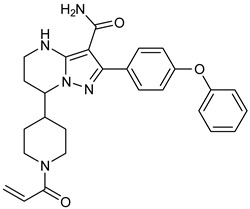	Tetrahydropyrazolopyrimidine 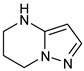	Mantle cell lymphoma
Tirabrutinib 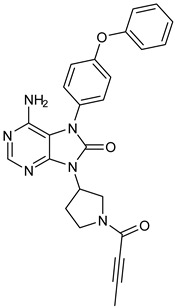	Pyrrolidinyl purine 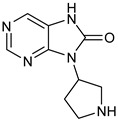	Recurrent or refractory primary central nervous system lymphoma, Waldenström’s macroglobulinemia, and lymphoplasmacytic lymphoma
Orelabrutinib 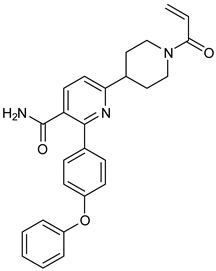	Piperidinyl pyridine 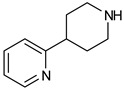	Mantle cell lymphoma, chronic lymphocytic leukemia, and small lymphocytic lymphoma
Remibrutinib 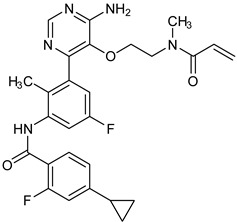	Pyrimidine 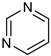	Autoimmune diseases

**Table 2 pharmaceuticals-16-00400-t002:** Non-covalent inhibitors of BTK under clinical trials.

Name	Heterocyclic Scaffold	Use
Vecabrutinib (SNS062) 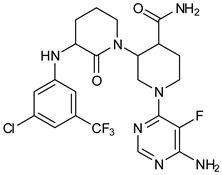	Pyrimidinylpiperidinyl piperidine 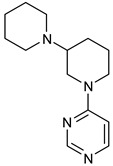	B-lymphoid cancers
Fenebrutinib (GDC0853) 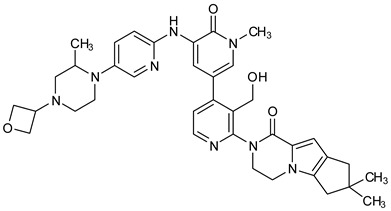	Piperazinylpyridine 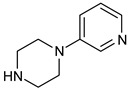	Inflammatory arthritis
Pirtobrutinib (LOXO305) 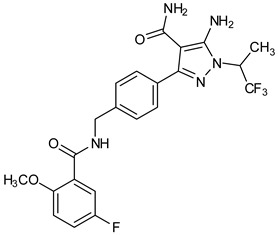	Pyrazole 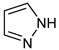	Chronic lymphocytic leukemia
Nemtabrutinib (ARQ531, MK1026) 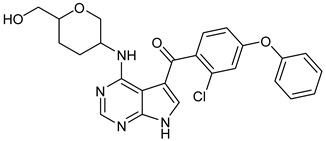	Pyrrolopyrimidine 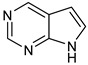	Chronic lymphocytic leukemia, Waldenström macroglobulinemia, mantle cell lymphoma, follicular lymphoma, marginal zone lymphoma, diffuse large B-cell lymphoma
RN-486 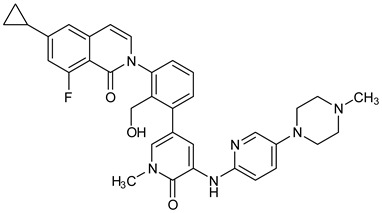	Isoquinoline 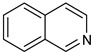	Rheumatoid arthritis and systemic lupus erythematosus
GNE-431 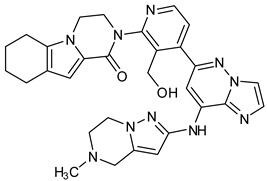	Pyrazinoindole 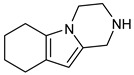	Acquired resistance to ibrutinib by mutation of Cys481 or Thr474
BMS935177 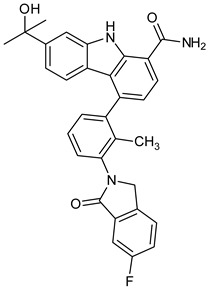	Carbazole 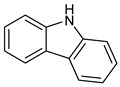	Inflammatory and autoimmune disease
Rilzabrutinib (covalent reversible inhibitor) 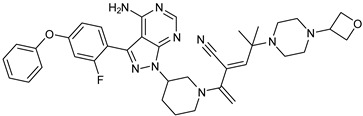	Pyrazolopyrimidine— 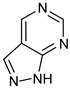	Immune thrombocytopenia, pemphigus vulgaris

**Table 3 pharmaceuticals-16-00400-t003:** PDB data of recent 3D structures of BTK-kinase domain with its complementary ligands.

PDB ID	Ligand ID/Scaffold	Under Clinical/Preclinical Trials	Nature of Ligand Binding
7N5Y	TAK020/Pyrrolidinyl-isoquinolinyl-triazole	Rheumatoid arthritis	Covalent
7KXQ	X9Y/Imidazopyridine	B-cell malignancies	Non-covalent
6W07	BIIB068/Pyrazolyl-pyrimidinyl-azetidine	Systemic lupus erythematosus	Non-covalent
7R60	2IE/Piperidinylpyridine	Autoimmune disorders	Covalent
7LTZ	BIIB091/Pyrazolyl-pyrimidinyl-azetidinylltriazole	Multiple sclerosis	Non-covalent
6XE4	V1G/Pthalazinyl-bipyridine	Rheumatoid arthritis	Non-covalent
6X3P	L-005298385/Imidazopyrazine	Rheumatoid arthritis	Non-covalent
6TFP	LOU064/Pyrimidine	Chronic urticaria, Sjogren’s syndrome	Covalent

## Data Availability

Data sharing not applicable.

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
