# Peer review of "Structural Complementarity of Bruton’s Tyrosine Kinase and Its Inhibitors for Implication in B-Cell Malignancies and Autoimmune Diseases"

_pharmaceuticals, 2023, doi:10.3390/ph16030400_

Round 1

Reviewer 1 Report

Dear Editor,

This MS is described " Structural complementarity of Bruton's tyrosine kinase and its inhibitors for implication in B cell malignancies and autoimmune diseases". However, the novelty of this article is not clear. What is the difference between this article and previously published articles (for example" Targeting Bruton’s tyrosine kinase in B cell malignancies / or/ Inhibitors targeting Bruton’s tyrosine kinase in cancers: drug development advances /or/ Role of Bruton’s tyrosine kinase in B cells and malignancies/….)?

Reviewer 2 Report

Dear Author

Thank you very much for writing a good review.

I hope you will refer to a few things and write your opinion.

1.If a particular author's opinion is not expressed, it would be good to include the PH-HH domain in Figure 2 to help readers understand. It would be better to match the contents with Figure 1 as much as possible.

2. The author summarized the inhibitors of BTK in a table. This helps to organize the results of recent research to the reader. I understand that you have composed the contents separately because of the importance of non-covalent inhibitors. My sugestions are

- If possible, please organize covalent inhibitors currently under clinical trial in a table together with the table of approved covalent inhibitors.

- The 2021 BMC review lists substances such as ARQ 531 and LOXO-305 for non-covalent inhibitors in B cell. Since your review didn't mention this material, I'd like to have some clarification on whether it's simply missing or terminated. If it is terminated and omitted from the list, Vecabrutinip is also terminated.

3. This is a fairly minor opinion, but please express the left cartoon of Figures 6 and 10 with better visibility.

Reviewer 3 Report

Comments and Suggestions for Authors

This is a good and on-time review.

Minor points

Please make sure the text is readable and of good quality in all figures. Also, enhance the quality of figures.

How it advances the current field.

Why it is important and timely review.

What are the shortcomings in the current field and how it should be addressed?

How was the literature searched and screened?

What method was used to draw the figures?

Reviewer 4 Report

Title: Structural Complementarity of Bruton's Tyrosine Kinase and its Inhibitors for Implication in B-cell Malignancies and Autoimmune Diseases

Manuscript ID: pharmaceuticals-2182012

Najmi et al have reviewed on covalent and non-covalent inhibitors of the BTK kinase domain revealing the complementary conformational alterations in the whole structure of BTK which are required to enhance the successful design and development of clinically effective drugs for implication in B-cell malignancies and autoimmune diseases.

.

Comments:

  1. Review has been well written but referencing part has major flaws. There is no reference number given. So, it’s very hard to track the cited articles and also, they are not in sequences. Also, the formatting of referencing is also not as per the journal requirement.
  2. Plagiarism has been also found 38% which is also high. So, it’s better to modify little bit.
  3. Image resolutions are not that good. Please improve image quality.
  4. In Tables, please include chemical structures of each drug and also for the mentioned heterocycles.
  5. In page 10, the homo sapiens should be italics.
  6. In page 13, positive charge on H and OH should be in superscript.
  7. Fig 6A is not very clear because of color coordination.
  8. On page 15, author has written ‘The terminal hydrophobic rings may be heterocyclic triazole or alkyl/fluoro substituted phenyl rings which can make the compounds selective to BTK’- Explain why?
  9. Please include citation for this line ‘The inhibitor chemistry 555 must be complementary to BTK in shape and size and is accomplished by the bulky het- 556 erocyclic core adjoining lipophilic rings and saturated heterocyclic spacer’.
  10. Please include the structure of fenebrutinib in figure 8 otherwise it is hard to understand the discussion part 8.2.
  11. The spelling of fluoro is wrong in page 18.
  12. The chemdraw structure in figure 10 B is not clear and stretched.
